# Stroke among cancer patients

Nicholas G. Zaorsky [1,2]*, Ying Zhang[2], Leila T. Tchelebi[1], Heath B. Mackley[1], Vernon M. Chinchilli[2] & Brad E. Zacharia[3]

We identify cancer patients at highest risk of fatal stroke. This is a population-based study using nationally representative data from the Surveillance, Epidemiology, and End Results program, 1992-2015. Among 7,529,481 cancer patients, 80,513 died of fatal stroke (with 262,461 person-years at risk); the rate of fatal stroke was 21.64 per 100,000-person years, and the standardized mortality ratio (SMR) of fatal stroke was 2.17 (95% CI, 2.15, 2.19). Patients with cancer of the prostate, breast, and colorectum contribute to the plurality of cancer patients dying of fatal stroke. Brain and gastrointestinal cancer patients had the highest SMRs (>2-5) through the follow up period. Among those diagnosed at <40 years of age, the plurality of strokes occurs in patients treated for brain tumors and lymphomas; if >40, from cancers of the prostate, breast, and colorectum. For almost all cancers survivors, the risk of stroke increases with time.

[1] Department of Radiation Oncology, Penn State Cancer Institute, Hershey, PA, USA. [2] Department of Public Health Sciences, Penn State College of Medicine, Hershey, PA, USA. [3] Department of Neurosurgery, Penn State College of Medicine, Hershey, PA, USA. *email: nicholaszaorsky@gmail.com

Cancer is the leading cause of death in the United States, and the third leading cause of death around the world[1]. Stroke is the fifth leading cause of death in the United States[2]. Cancer and stroke may occur separately in a patient, or cancer may lead to stroke via hypercoagulability, non-bacterial thrombotic endocarditis, direct tumor compression of blood vessels, or from therapy[3]. As the survival rates of cancer patients continue to increase, it will become crucial to identify cancer survivors at elevated risk of stroke.

The American Heart Association and American Stroke Association provide guidelines for the prevention of stroke in patients with stroke and transient ischemic attack[4] and for early management of patients with acute ischemic stroke[5]. The National Comprehensive Cancer Network provides survivorship guidelines after therapy for cancer, with the goal of preventing long-term morbidity and mortality[6]. As of 2019, these organizations offer relatively limited guidelines for stroke prevention, identification, or management specifically in cancer patients. Thus, there is currently no resource to assist clinicians, including primary care physicians, oncologists, neurologists, neurosurgeons, and cardiologists, in identifying cancer patients at highest risk of stroke. One strategy of stroke prevention in cancer patients is to identify and target subgroups at greatest risk of stroke.

The purpose of the current work is to present an analysis of the risk of stroke among cancer patients. Our objectives are to identify cancer patients at highest risk of fatal stroke compared to (1) the general population, and (2) other cancer patients. This work may be used clinically by physicians in the creation of survivorship programs to mitigate the risk of stroke among cancer patients.

## Results

**Cancer patient risk of fatal stroke vs general population.** A total of 7,529,481 cancer patients were included in the analysis; of these, 80,513 (1.1%) died of a stroke. Among all cancer patients, the rate of stroke per 100,000-person-years was 21.64, and the SMR of stroke was 2.17 (95% CI, 2.15, 2.19, relative risk $p < 0.0001$).

Table 1 shows the characteristics of all cancer patients included as well as those who died of stroke vs all cancer patients. Males and females were both equally likely to die of stroke, 51.6% vs 48.4%. Patients who were diagnosed at a younger age had a higher SMR for stroke, and the SMRs gradually declined as patients were diagnosed at a later age; those <39-years had an SMR of 81.09 (95% CI 61.42, 105.07, relative risk $p < 0.0001$) vs >80-year-olds had an SMR of 1.84 (95% CI 1.81, 1.86, relative risk $p < 0.0001$). Although there were only 8,076 (10.0%) patients with metastatic/distant disease at diagnosis, these patients had the highest SMR of death from stroke, 5.06 (95% CI 4.88, 5.25, relative risk $p < 0.0001$). Those diagnosed 1992–2000 had an SMR of 1.51 (95% CI 1.47, 1.54, relative risk $p < 0.0001$), and those diagnosed 2011–2015 had an SMR of 5.25 (95% CI 5.14, 5.35, relative risk $p < 0.0001$).

---

**Table 1 Standardized mortality ratios of stroke among cancer patients**

|  | Total[a] | Strokes[a] | Strokes per 10,000 person-years[a] | SMR (95% CI)[b] |
|---|---|---|---|---|
| **Age group** |  |  |  |  |
| ≤39 | 450,691 (6.0%) | 300 (0.4%) | 0.89 | 81.09 (61.42, 105.07) |
| 40–49 | 669,634 (8.9%) | 1107 (1.4%) | 2.42 | 33.73 (29.39, 38.53) |
| 50–59 | 1,377,815 (18.3%) | 4408 (5.5%) | 5.42 | 14.69 (13.75, 15.67) |
| 60–69 | 1,928,342 (25.6%) | 13,857 (17.2%) | 13.71 | 5.52 (5.32, 5.72) |
| 70–79 | 1,864,031 (24.8%) | 30,939 (38.4%) | 38.63 | 2.55 (2.50, 2.60) |
| 80+ | 1,238,968 (16.5%) | 29,902 (37.1%) | 98.85 | 1.84 (1.81, 1.86) |
| **Sex** |  |  |  |  |
| Female | 3,661,011 (48.6%) | 39,002 (48.4%) | 20.84 | 2.28 (2.25, 2.31) |
| Male | 3,868,470 (51.4%) | 41,511 (51.6%) | 22.44 | 2.08 (2.05, 2.11) |
| **Race** |  |  |  |  |
| White | 6,186,237 (82.2%) | 67,509 (83.8%) | 21.74 | 2.05 (2.02, 2.07) |
| Black | 770,801 (10.2%) | 7997 (9.9%) | 23.80 | 2.74 (2.65, 2.83) |
| Other | 499,751 (6.6%) | 4828 (6.0%) | 19.81 | 3.37 (3.25, 3.49) |
| Unknown | 72,692 (1.0%) | 179 (0.2%) | 5.03 | 0.00 (0.00, 0.00) |
| **Marital status** |  |  |  |  |
| Married | 4,092,712 (54.4%) | 40,083 (49.8%) | 17.83 | 0.91 (0.89, 0.92) |
| Unmarried | 2,895,946 (38.5%) | 34,581 (43.0%) | 28.64 | 2.32 (2.29, 2.36) |
| Unknown | 540,823 (7.2%) | 5849 (7.3%) | 22.02 | 2.08 (2.00, 2.17) |
| **Stage** |  |  |  |  |
| Distant | 1,526,068 (20.3%) | 8076 (10.0%) | 28.27 | 5.06 (4.88, 5.25) |
| Regional | 2,309,633 (30.7%) | 25,503 (31.7%) | 19.24 | 2.20 (2.16, 2.24) |
| Localized | 2,508,615 (33.3%) | 32,796 (40.7%) | 19.92 | 1.88 (1.84, 1.93) |
| Unstaged/unknown | 1,185,165 (15.7%) | 14,138 (17.6%) | 30.48 | 1.99 (1.94, 2.04) |
| **Year of diagnosis** |  |  |  |  |
| 1992–2000 | 1,624,977 (21.6%) | 35,090 (43.6%) | 27.34 | 1.51 (1.47, 1.54) |
| 2001–2005 | 1,824,980 (24.2%) | 23,752 (29.5%) | 20.16 | 1.75 (1.72, 1.79) |
| 2006–2010 | 1,992,271 (26.5%) | 15,349 (19.1%) | 17.05 | 2.10 (2.06, 2.15) |
| 2011–2015 | 2,087,253 (27.7%) | 6322 (7.9%) | 17.60 | 5.25 (5.14, 5.35) |
| **Surgery** |  |  |  |  |
| Yes | 4,315,322 (57.3%) | 49,965 (62.1%) | 18.14 | 2.04 (2.01, 2.07) |
| No | 3,062,213 (40.7%) | 28,958 (36.0%) | 30.65 | 2.45 (2.41, 2.50) |
| Unknown | 151,946 (2.0%) | 1590 (2.0%) | 70.52 | 3.81 (3.25, 4.44) |
| All patients | 7,529,481 | 80,513 (1.1%) | 21.64 | 2.17 (2.15, 2.19) |

[a]Database "SEER 18 Regs Research Data + Hurricane Katrina Impacted Louisiana Cases, Nov 2017 Sub (1973–2015 varying)" was used
[b]Database "Incidence - SEER 13 Regs excluding AK Research Data, Nov 2017 Sub (1992–2015) for SMRs" was used; exact method was used to calculate 95% CI

---

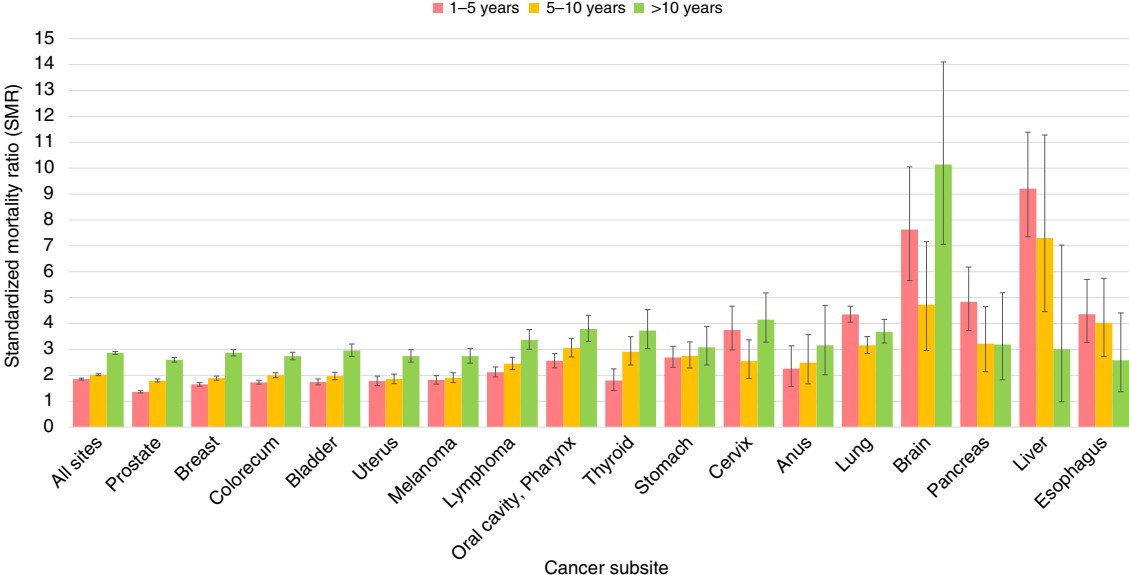

**Fig. 1** Standardized mortality ratios (SMRs) of fatal stroke among cancer patients by subsite. The *y*-axis depicts the SMR with 95% CI, and the *x*-axis depicts the disease site. Different time periods after diagnosis (<1 year vs 5–10 years vs >10 years) are shown in different colors. The risk of stroke among cancer patients is two times that of the general population and rises with longer follow-up time. Certain cancer patients have relatively high SMR from stroke in the first year after diagnosis (e.g., brain, with SMR 7). Error bars represent 95% CI by subsite. Source data are provided as a Source Data file

Figure 1 shows SMRs of stroke among cancer patients by subsite. Overall, the risk of stroke for cancer patients is more than twice that of the general population (for all sites, the SMR at 1–5 years after diagnosis is 1.87, 95% CI 1.81, 1.89; relative risk *p* < 0.0001), and this risk increases with follow-up time (for all sites, the SMR at >10 years after diagnosis is 2.87, 95% CI 2.81, 2.92; relative risk *p* < 0.0001). Certain cancer patients have relatively high SMR from stroke in the first year after diagnosis. For example, upper gastrointestinal malignancies (pancreas, liver, and esophagus) have SMRs of 3–5 in the first year of diagnosis (relative risk *p* < 0.0001). Similarly, brain tumor patients have an SMR of 7.63 (95% CI 5.66, 10.5, relative risk *p* < 0.0001) in the first 5 years following diagnosis, and this SMR remains elevated throughout all follow-up times. For reference, the SMR for all cancer subsites available through SEER is available in the Source Data file.

**Cancer patient risk of fatal stroke vs other cancer patients.** Table 2 (left panel) shows the ORs of patients who died of stroke, stratified by subgroup. Patients older than 80 years of age have a stroke OR of 43.64 (95% CI 38.94, 48.90) compared to those <39 years of age. Blacks have an OR of 1.26 (95% CI 1.23, 1.29) compared to whites. Patients with localized disease had a higher OR of stroke compared to those with distant metastases, OR of 2.56 (95% CI 2.49, 2.63).

Figure 2 shows the cancer patients who died from stroke as a function of age group. Table 2 (Cox proportional hazards model in right panel) shows the HRs of patients who died of stroke, stratified by subgroup, complementing the results of Fig. 2. Figure 3 shows fatal strokes among cancer patients as a function of age group. Relatively few patients <40 years of age died of stroke, in part because most cancers are diagnosed in the elderly. Among patients diagnosed at age <40, the plurality of stroke occurs in patients with brain tumors and lymphomas. In contrast, among patients diagnosed at age >40, the plurality of strokes occurs in patients diagnosed with prostate, breast, and colorectal cancer. The relative risk of stroke is dramatically higher in the elderly: HR 80+ year-olds, vs those ≤39 year-olds is 169.73 (95% CI 134.64, 169.73, Type III *p* < 0.0001).

Supplementary Table 1 shows the SMRs of cancer patients diagnosed at ages 0–19 years who died of stroke. This subanalysis was included to further explore the high SMR observed in the <39 age group as presented in Table 1. Notably, there are few person-years at risk per cancer subsite due to the rarity of the combination of cancer and fatal stroke in pediatric patients.

We performed a subgroup analysis of patients with only brain tumors, to investigate if the risk of stroke was substantially higher in these patients. Among patients with brain tumors, there were 525 events of 99,434 patients, with a median time to stoke of 5 months; in contrast, among other patients, there were 79,988 events, out of 743,0051 patients, with a median time to fatal stroke of 52 months. In the adjusted Cox regression, the hazard ratio of fatal stroke in brain tumor patients vs all others was 3.085 (95% CI 2.824, 3.369, Type III *p* < 0.0001). Similarly, Supplementary Figure 1, shows SMRs in subsites of the head and neck; there was no single subsite that appeared to be a higher risk.

**Discussion**
We present a contemporary analysis of risk of fatal stroke among more than 7.5 million cancer patients and report that stroke risk varies as a function of disease site, age, gender, marital status, and time after diagnosis. The risk of stroke among cancer patients is two times that of the general population and rises with longer follow-up time. The relative risk of fatal stroke, vs the general population, is highest in those with cancers of the brain and gastrointestinal tract. The plurality of strokes occurs in patients >40 years of age with cancers of the prostate, breast, and colorectum. Patients of any age diagnosed with brain tumors and lymphomas are at risk for stroke throughout life.

Most cancer patients now die of non-cancer causes[7]. The results of the current work suggest that stroke-prevention strategies may be aimed at patients treated for brain tumors and lymphomas (particularly children) and older patients (i.e., >40 year-olds) diagnosed with cancers of the prostate, breast, and colorectum. Though relatively less common, patients with cancers of the gastrointestinal tract (especially the pancreas, liver, esophagus) are at a relatively high risk to die of stroke at any time after diagnosis (SMRs 3-10). We encourage individual guideline

**Table 2 Odds ratios and hazard ratios of stroke among cancer patients**

| | Logistic regression model | | | Cox proportional hazards model | | |
|---|---|---|---|---|---|---|
| | Odds ratio | 95% CI | P-value[a] | Hazard ratio | 95% CI | P-value[a] |
| Age Group | | | <.0001 | | | <.0001 |
| ≤39 | 1.00 | – | | 1.00 | – | |
| 40–49 | 2.49 | (2.19, 2.83) | | 3.65 | (2.82, 3.65) | |
| 50–59 | 5.42 | (4.82, 6.10) | | 8.65 | (6.82, 8.65) | |
| 60–69 | 12.49 | (11.14, 14.00) | | 22.19 | (17.58, 22.19) | |
| 70–79 | 27.51 | (24.55, 30.83) | | 63.30 | (50.23, 63.3) | |
| 80+ | 43.64 | (38.94, 48.90) | | 169.73 | (134.64, 169.73) | |
| Sex | | | <.0001 | | | 0.0240 |
| Female | 1.00 | – | | 1.00 | | |
| Male | 1.04 | (1.02, 1.05) | | 1.04 | (1, 1.04) | |
| Race | | | <.0001 | | | <.0001 |
| White | 1.00 | – | | 1.00 | | |
| Black | 1.26 | (1.23, 1.29) | | 1.38 | (1.32, 1.38) | |
| Other | 1.06 | (1.03, 1.10) | | 1.07 | (1.01, 1.07) | |
| Unknown | 0.39 | (0.34, 0.45) | | 0.38 | (0.28, 0.38) | |
| Marital status | | | <.0001 | | | <.0001 |
| Married | 1.00 | – | | 1.00 | | |
| Unmarried | 1.07 | (1.05, 1.09) | | 1.33 | (1.29, 1.33) | |
| Unknown | 1.20 | (1.17, 1.24) | | 1.11 | (1.05, 1.11) | |
| Stage | | | <.0001 | | | <.0001 |
| Distant | 1.00 | – | | 1.00 | | |
| Regional | 2.10 | (2.05, 2.16) | | 0.76 | (0.72, 0.76) | |
| Localized | 2.56 | (2.49, 2.63) | | 0.85 | (0.8, 0.85) | |
| Unstaged/unknown | 1.94 | (1.89, 2.00) | | 0.98 | (0.93, 0.98) | |
| Year of diagnosis | | | <.0001 | | | <.0001 |
| 1992–2000 | 1.00 | – | | 1.00 | | |
| 2001–2005 | 0.61 | (0.60, 0.62) | | 0.78 | (0.76, 0.78) | |
| 2006–2010 | 0.36 | (0.36, 0.37) | | 0.68 | (0.65, 0.68) | |
| 2011–2015 | 0.14 | (0.14, 0.15) | | 0.61 | (0.57, 0.61) | |
| Surgery | | | <.0001 | | | <.0001 |
| Yes | 1.00 | – | | 1.00 | | |
| No | 0.81 | (0.80, 0.83) | | 1.24 | (1.2, 1.24) | |
| Unknown | 0.69 | (0.65, 0.72) | | 1.39 | (1.19, 1.39) | |

[a]Type III

and survivorship committees to incorporate these data into their stroke-prevention statements. We recommend that providers follow the evolving guidelines for monitoring distress and stroke prevention from the National Comprehensive Cancer Network, American Heart Association and American Stroke Association[4–6].

Relatively few studies have examined the risk of stroke among cancer patients, and the current analysis is the largest of its kind (Table 3)[8–14]. Notably, in other studies, the number of patients is relative small (typically <100 vs >80,000 in the current work), the types of patients included are much more limited (most other studies only focus on lung, prostate, colorectal, or breast cancer patients), and the conclusions of other works are therefore limited. Similar to previous analyses, we found that lung, prostate, breast, and colorectal patients experience the plurality of strokes. Although the current analysis does not include patient comorbidities or biomarkers, other studies suggest that D-dimer levels and classic risk factors for stroke (e.g., hypertension) put patients at greatest risk.

Although it is estimated that 12% of stroke patients have an occult malignancy[10], few studies evaluate cancer patients at an elevated risk for stroke. Cestari et al.[14] assessed the incidence and type of strokes in cancer patients at Memorial Sloan Kettering Cancer Center; among 96 patients, they report that lung cancer (30%) was the most common primary tumor followed by brain and prostate cancer (9% each). In the current analysis, we similarly noted a high risk of stroke in prostate and brain tumor

patients; we unfortunately cannot comment on the embolic vs non-embolic nature of stroke because of the limitations of the SEER database.

The risk factors for stroke in cancer patients are under investigation. Nguyen and DeAngelis[15] systematically reviewed the literature of types of cancer patients with comorbid stroke, including data from the American Cancer Society, the Danish Hospital Discharge Registry[12], University of Massachusetts Medical Center[13], and Memorial Sloan Kettering Cancer Center[14]. The authors report that patients with cancer are subject to the same stroke risk factors as the general population, and atherosclerosis remains the most common cause of stroke in cancer patients. Further, they note that if stroke in cancer patients was caused by the same pathophysiologic mechanisms as in the general population, the distribution of stroke should be identical to the population at large, and there would be a distribution of primary neoplasms proportional to the most common cancers (i.e., lung, breast, and prostate). In their review, there was a relatively wide variability of stroke among tumor types, e.g., data from Denmark[12] and Massachusetts[13] revealed comorbid stroke in 28 and 11% of gastrointestinal cancer patients, respectively.

Jagsi et al.[9] report that for early stage breast cancer patients, only age and hypertension remained significant predictors of stroke and transient ischemic attack, after adjusting for coronary artery disease, atrial fibrillation, and supraclavicular radiation therapy. Craniospinal radiation therapy increases risk of stroke in pediatric cancer patients, purportedly due to microvascular

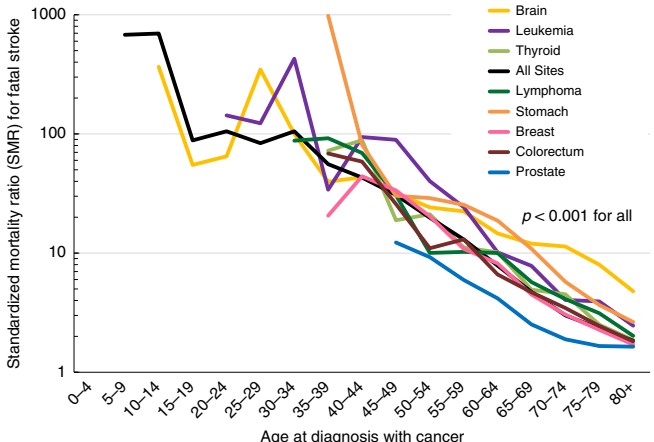

**Fig. 2** Standardized mortality ratios (SMRs) of fatal stroke as a function of age of diagnosis. The y-axis depicts the SMR, and the x-axis depicts the age at diagnosis with cancer. SMRs compare the risk of death from stroke among a cancer subsite vs the general population, adjusted sex and race, within a particular age subgroup. Cancers are shown in different colors; for the purposes of this figure, key cancers were selected because of their high incidence and prevalence overall (e.g., prostate, breast, colorectum) and because of their relatively high incidence in pediatric populations (e.g., brain, leukemia); this was done so that SMRs between adult and pediatric populations may be juxtaposed. For pediatric patients, the population is enriched with brain tumors, and these contribute to the majority of person-years at risk. Children diagnosed with brain tumors are at an exceedingly high risk to die of stroke for the remainder of their life (SMRs > 100, p-values < 0.001). Adolescents and young adults who are diagnosed with leukemia are similarly at a high risk of death from fatal stroke (SMRs > 100, p-values < 0.001). Since most cancers are diagnosed in adults and the elderly, SMRs for the majority of other cancers are not plotted until age 40 and over. In general, the younger a patient's age of diagnosis, the higher the SMR that the patient will die of stroke through their life. Source data are provided as a Source Data file

damage[16], which may be exacerbated by comorbidities such as diabetes[17]. Overall, the incidence of ischemic and non-ischemic stroke among cancer and non-cancer patients appears to be relatively similar, at 85 and 15%, respectively[10].

Several pathways for increased risk of stroke in cancer patients have been proposed, and there are several cancer-specific types and causes of stroke in cancer patients[18]. Nguyen and Deangelis[3] describe that cancer may lead to stroke via several mechanisms. First, certain cancers cause occlusive disease from emboli, compression, or meningeal extension of tumor. Tumor dissemination into the leptomeningeal space can lead to vascular compromise. Patients with leukemia and elevated leukocyte counts may develop intravascular leukostasis, leading to hemorrhagic infarct. Brain tumor metastases may also cause hemorrhage, and this is more common in cancers of the kidneys, thyroid, germ cells, melanoma, and choriocarcinoma. Second, coagulopathies, including non-bacterial thrombotic endocarditis (NBTE), may cause stroke. NBTE, or marantic endocarditis, is characterized by the presence of relatively acellular aggregates of fibrin and platelets attached to normal heart values. Third, stroke may occur from therapy, such as radiation therapy-induced atherosclerosis, drug-induced thrombocytopenia, and hypercoagulability.

Our work has limitations. The overall number of deaths from stroke was relatively limited overall (1% of cancer patients), and more detailed analyses on risk factors could not be performed. Treatment paradigms have changed since the 1990s; for example, Hodgkin lymphoma patients are now treated with limited chemotherapy, and possibly a relatively low dose of very targeted radiation[19], which would decrease atherosclerosis and risk of stroke. Additionally, patients having death events in earlier years (i.e., 1990s) have limited follow-up and less time at risk (≤10 years) than some patients with events in more recent years. This may have resulted in an overestimate of SMRs for individuals diagnosed between 2011-2014, compared to those diagnosed before 2000. Similarly, patients diagnosed in recent years have short follow-up and lower chance of death from any cause.

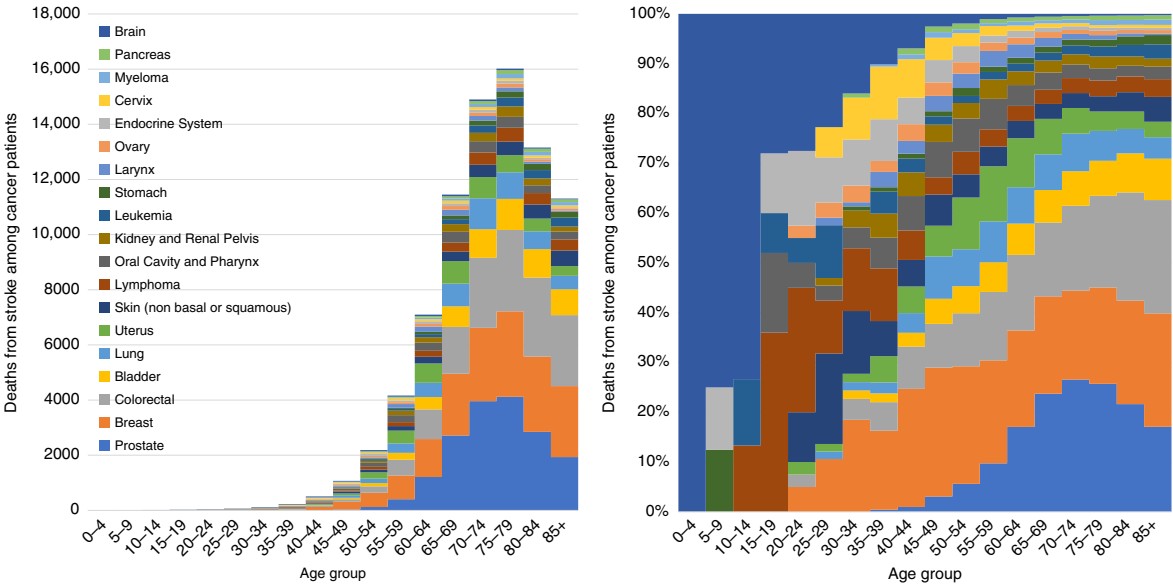

**Fig. 3** Fatal stroke among cancer patients as a function of age group. **a** The y-axis depicts the absolute number of strokes and the x-axis depicts the age group at time of diagnosis. The colors depict the disease sites. The majority of strokes are in patients diagnosed at an older age (i.e., 40–90-year-olds), and the plurality of strokes s occurs in patients diagnosed with prostate, breast, and colorectal cancer. **b** The y-axis depicts the relative number of strokes compared to all cancer patients, and the x-axis depicts the age group at time of diagnosis. The colors depict the disease sites. For children, adolescents, and young adults (i.e., <40 years old), the plurality of strokes is seen in brain tumor patients. In contrast, among older adults (i.e., >40 years old) the plurality of strokes occurs in patients with cancer of the prostate, breast, and colorectal cancer. Source data are provided as a Source Data file

**Table 3 Literature review on the analyses of stroke among cancer patients**

| Study | Type of article/ location | n | Principal cancers included | Follow-up time | Findings |
|---|---|---|---|---|---|
| Zaorsky et al. (Current study) | USA | 7,529,481 cancer patients, 80,513 died of stroke (larger than all other articles below combined) | All. Notably, includes cancers of head and neck, heme system, pediatrics, GI, brain (which are typically not included in studies below); provides risk vs general and cancer populations | >20 years for cohort follow-up, >260,000 person-years at risk for standardized incidence ratios. Median time of death for fatal stroke 5 months for brain tumor patients vs 52 months for non-brain patients. | Brain and GI cancer patients had the highest SMRs (>2–5) through the follow-up period. Among those diagnosed at <40 years of age, plurality of strokes occurs in patients treated for brain tumors and lymphomas; if >40, from cancers of the prostate, breast, and colorectum. For almost all cancer survivors, the risk of stroke increases with time. |
| Kim et al.[8] | Korea, 6 centers | 161 cancer patients who had stroke | Mostly lung, gastric, colorectal | None, no survival analyses possible | Patients are at risk for cryptogenic and conventional stroke. D-dimer levels are higher in stroke patients (odds ratio 10–11). |
| Jagsi et al.[9] | Ann Arbor, Michigan, USA | 820 early breast cancer patients, 35 had stroke | Early breast only | 6.8 years | SMR of stroke is 1.7–2.8 among breast cancer patients. Hypertension and age are predictors. |
| Stefan et al., 2009[10] | Vienna, Austria | 1274 stroke patients, 12% of these had cancer | Mostly breast, prostate, colorectal | None, no survival analyses possible | Cerebrovascular risk factors do not significantly vary between cancer and non-cancer patients. |
| Zhang et al.[11] | Australia | 69 stroke patients with cancer at one hospital | Mostly prostate, lymphoma | None, no survival analyses possible | In cancer patients, trend toward higher risk of intracerebral hemorrhage and higher partial thromboplastin time. |
| Lindvig et al., 1990[12] | Denmark | 113,732 stroke patients, 5151 had cancer | Mostly lung | 2.4 years | Overall, more cancer was expected than observed. No risk factor between stroke and gastric cancer |
| Chaturvedi et al.[13] | Massachusetts, MA, USA | 33 patients with cancer who had stroke, representing 3.5% of admissions to hospital | Mostly GYN, genitourinary, gastrointestinal | 9 months | Recurrent cerebral ischemic events were noted in only 6% of patients |
| Cestari et al.[14] | NYC, NY, USA | 96 patients with cancer who had stroke | Mostly lung, breast, prostate | After stroke median survival was 5 months | 54% of patients had embolic strokes, partially due to hypercoagulability, with 11/12 patients having elevated D-dimer levels |

While SEER data were beneficial to use for this sort of analysis, it is not without limitations. SEER contains basic information regarding diagnosis, cause of death, and first treatment type. It does not contain information regarding stroke subtype, comorbidities that may increase a patient's risk of stroke (e.g., smoking, hypertension), biomarkers (e.g., prothrombin time, D-dimer levels), or the full extent of their treatment. Notably, no other databases published to date contain these covariates (Table 3), and a nationally representative database containing this much information does not exist. We recognize these limitations and do not attempt to extrapolate our findings to specific subpopulations beyond the variables included in the analysis. Our findings are valuable in better understanding stroke risk in cancer patients.

In the current work, we note that the incidence of fatal stroke among cancer patients is relatively low at 1%. In other analyses (Table 3), the rate of non-fatal strokes has been reported at ~5% among cancer patients, and even strokes that do not cause death may be debilitating. However, since there is a gradient with stroke diagnoses (e.g., transient ischemic attacks that leave no symptoms), we focus on only the most significant strokes (i.e., those

that cause death), and we believe the results may be extrapolated to other cancer patients. A potential limitation of this work may be that cancer patients do not have increased risk of stroke but might have increased fatalities owing to poorer health due to cancer.

Further, there is a risk of bias and misclassification of cause of death (including stroke) in the SEER[20,21], e.g., for lymphoma histologies[22], and receipt of radiation therapy[23]. Thus, we are unable to characterize misclassification of stroke in the current work. It is possible that certain strokes were caused by brain metastases or primary tumors themselves, rather than ischemia. Stroke is also a common accompanying morbidity of brain tumor surgery, and it is possible that the data do not separate perioperative strokes from others.

The changes in stroke diagnosis and treatment and cancer diagnosis and treatment have changed from 1992 to 2015. There may be some variability in the relative risk of fatal stroke among patients diagnosed in earlier years vs later years. Thus, we have performed additional analyses to address this concern, and we note that the relative risk of death from fatal stroke (as measured

with the standardized mortality ratio, SMR) for 1992–2002 and 2005–2015, at 1.58 (95% CI 1.55, 1.61) vs 2.95 (95% 2.91, 2.99). It is difficult to compare SMRs to each other, since their reference populations (in this case the general population in the US) differ; nonetheless, we hypothesize that the SMRs from more recent years may be higher than those seen in the 1990s because in the United States there has been a trend to detect and treat more low risk cancers (e.g., prostate, breast, and colorectal); these patients likely will never die of their primary cancers and are therefore at higher risk to die of other causes. We provide the full dataset for these time periods in the Source Data file.

The results of the current work suggest the risk of stroke among cancer patients is twice that of the general population and rises with longer follow-up time. The relative risk of stroke, vs general population, is highest in those with cancers of the brain and the upper gastrointestinal tract. The plurality of strokes occurs in patients >40 years of age with cancers of the prostate, breast, and colorectum. Patients of any age diagnosed with brain tumors are at risk for stroke throughout life.

## Methods

**Data acquisition**. Patients with invasive cancer, diagnosed between 1992 and 2015, were abstracted from the National Cancer Institute's Surveillance, Epidemiology, and End Results (SEER) program[24,25]. The overview and limitations of the database and the methods are described in the Supplementary Notes[26–29]. SEER is a network of population-based incident tumor registries from geographically distinct regions in the US, covering 28% of the US population, including incidence, survival, and surgical treatment[24,25]. For the current analysis, the SEER 18 registry was used. The SEER registry includes data on sex, age at diagnosis, race, marital status, and year of diagnosis. SEER does not code comorbidities, performance status, surgical pathology, doses, radiotherapy use, and chemotherapy agents. SEER*Stat 8.2.1 was used for analysis[24].

All patients with an invasive cancer diagnosis were included. Patients diagnosed only through autopsy or death certificate (<1.5% of patients) were excluded. Data were extracted for cancers with more than 100,000 person-years or more of survival time; thus, certain uncommon and aggressive cancers were excluded, including Kaposi's sarcoma, multiple myeloma, male breast cancers, and mesotheliomas. Leukemias and lymphomas were grouped for certain parts of the analysis, so they could be reported accurately. Time in SEER is measured in months, and the smallest nonzero value is 1 month, which was the minimum time to any event.

Mortality codes in SEER are assigned from death certificates, completed by the doctor caring for the patient at the time of demise. Stroke is defined by the American Stroke Association[4] as central nervous system infarction attributable to ischemia, based on neuropathological, neuroimaging, and/or clinical evidence of permanent injury. For the purposes of this study, patients were considered to have died of stroke if the death certificate stated: cerebrovascular accident, International Classification of Diseases 9 (ICD-9) code 434.11, or ICD-10 code I63.9. Unfortunately, further details of stroke are unavailable, e.g., if stroke was hemorrhagic vs embolic, if there was a ruptured aneurysm, if stroke was peri-operative.

Notably, SEER does not code comorbidities or diagnoses associated with stroke, including smoking status, presence of hypercoagulable disease, prior thromboembolic events, heart disease, prior stroke, diabetes, or cholesterol levels. The observed associations between cancer and stroke may be confounded by prothrombotic disorders, the use of medications, and patient lifestyle factors, but we are unable to control for these factors in the current work. These are limitations of the analysis and limit the interpretability of the results.

For objective 1, we calculated standardized mortality ratios (SMRs), which provide the relative risk of death for patients with cancer as compared to all US residents, stratified by cancer subgroup[24,30,31]. Although SEER data extend back to 1972, SMRs were calculated for the time period 1992–2015 because the identification and treatment of stroke has changed drastically over the decades, and we felt that the current time range would provide more contemporary outcomes. Data were characterized with SMRs adjusted by age, race, and sex to the US population over the same time. Five-year-aged categories were used for standardization using SEER*Stat 8.2.1 and Microsoft Excel 15.0.4 (Microsoft, Redmond, WA)[31–33]. There are two important caveats about SMRs: (1) SMRs cannot be compared to each other, since they compare the relative risk vs. the standard population, and the standard population may be different among groups; (2) SMRs and their confidence intervals depend on person-years at risk, and if the incidence of a cancer is low and the survival is limited (e.g., liver cancer in the US), the CI may be wide.

For objective 2, odds ratios (ORs) with 95% CIs were calculated based on the number of observed events per patient subgroup, also for the time period 1992–2015, which we thought would provide more contemporary outcomes. Further, the absolute and relative number of strokes per patient age group (at time

of diagnosis) were calculated. We also performed a survival analysis using a Cox proportional hazards model to calculate hazard ratios (HRs), with the survival time being from diagnosis until stroke, and non-stroke deaths plus living patients being censored. In the analyses for both objective 1 and objective 2 of this manuscript, patients who die of their cancer are no longer at risk for a fatal stroke, and this is considered in the standardized mortality ratios, odds ratios, and hazard ratios.

## Data availability

The data are provided in the SEER database, which is freely accessible to the public. The relevant session information, i.e., the user-submitted request, from in the current work and abbreviated dataset (from SEER) are provided in the Source Data file. The source data underlying all tables and figures are provided as a Source Data file.

We comply with all relevant ethical regulations. The datasets generated and analyzed during the current study are available in the SEER repository (https://seer.cancer.gov/seerstat/). These data are freely available via the National Cancer Institute SEER program, and thus the study was exempt from institutional review board review. There are no participants in the study, and thus there is no consent form.

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

## Acknowledgements

We are thankful for the insight of Dr. Alexander C Mamourian, MD, of the Department of Radiology, and Penn State College of Medicine, whose comments greatly improved the quality of this work.

## Author contributions

All authors had full access to all of the data in the study and take responsibility for the integrity of the data and the accuracy of the data analysis. Study concept and design: N.Z., Y.Z., B.Z. Acquisition, analysis, and interpretation of data: N.Z., Y.Z., V.C. Drafting of the manuscript: N.Z. Critical revision of the manuscript for important intellectual content: N. Z., Y.Z., L.T., H.M., V.C., B.Z. Statistical analysis: N.Z., Y.Z., V.C. Obtained funding: N/A. Administrative, technical, or material support: N.Z. Study supervision: N.Z.

## Competing interests

The authors declare no competing interests.
