## [Peer Review File · Nature Communications]

Reviewers' comments:

Reviewer #1 (Remarks to the Author):

The objective of trying to identify cancer patients at risk for stroke is an important one. However, the limitations of the data source considerably limit the ability to glean useful information along this objective. As noted by the authors, the SEER database is limited in terms of the information it contains. The important information missing are patient co-morbidities at baseline. It is likely that there are important overlapping risk factors for stroke and cancer such as diet, exercise, diabetes, and smoking. Hence, it is not clear that a comparison to the general population is that informative but what would be more informative is a comparison of stroke incidence of cancer patients to a population enriched for stroke risk factors. This would allow the identification of risk factors that are unique to cancer patients outside of the known stroke risk factors. In addition, the variables in the SEER database are relatively crude and primarily limited to baseline patient and disease characteristics and first treatment course for the cancer. This severely limits being able to identify the cancer patient subgroups that would benefit the most for heightened stroke prevention measures. This is important given that only about 1% of the cancer patients suffered a fatal stroke. Given this low incidence, it does not necessarily make sense to implement something for the entire population given only a small number of patients may benefit.

Specific items for consideration:

- Given that cancer is a given risk factor for cancer, is there any evidence that implementing stroke risk identification/prevention beyond what is currently being done would reduce fatal stroke incidence in this patient population? Specifically, is there evidence that cancer patients who are not at higher risk for stroke given other co-morbidities and lifestyle choices are at higher risk of stroke? The concern is that the higher observed SMR may reflect a population that has a higher burden of risk factors for stroke rather than a mechanism of the cancer.
- The authors have mentioned several limitations that make it difficult to interpret these results. These include (1) the changes in stroke diagnosis and treatment and cancer diagnosis and treatment over the 23 years included in this analysis (how do the authors explain the increasing SMR as the date of diagnosis is more recent?), (2) the inability to distinguish between hemorrhagic and ischemic stroke (given some cancer treatments increase the risk of major bleeds), (3) and confounding by lifestyle choices that may place a patient at increased risk for both stroke and cancer. How do the authors envision their findings being able to guide physicians in the care of cancer survivors beyond what they are already doing?
- It seems like there is a competing risks issue. In particular, patients who die early due to their cancer would no longer be at risk for a fatal stroke. Has this been considered?

- The increase in stroke in brain tumor patients may be the result of an increase in stroke due to brain surgery (a known risk factor). Is there any way to determine whether brain cancer patients had a shorter time to fatal stroke versus other types of cancer patients (i.e. has some type of time to event analysis been performed to try to understand this?)
- How reliable are causes of death in the SEER database? What is the validity of this outcome?
- It is noted that stroke rates differ by disease site, age, gender, marital status, and time after diagnosis. Are any of these factors new and would any of these impactfully influence how cancer survivors should be managed differently than what is already being done? It seems like these are fairly crude measures and would not be able to tailor cancer survivorship plans to result in meaningful decreases in fatal stroke incidence.
- A real concern is that one biggest risk factor for subsequent stroke may be the type of treatment a patient received. This has not been analyzed in this study. Have the authors tried to analyze this? Even if possible, there is an additional concern that the patients only have data available for their first line of treatment. Longer cancer survivors are more likely to have undergone several different treatments, which may partially explain that the patients who have survived longer (i.e. a longer time from diagnosis) have higher SMRs in general. It seems that in order to more finely identify cancer patients at higher risk of fatal stroke, the most important variables would be stroke risk factors (so the analysis can adjust for these to find new factors) and patient treatment. There is also concern that once diagnosed with cancer, patient lifestyles may change in a way that places them at higher risk of stroke. These changes may be disease site specific as a consequence of morbidities associated with the cancer and its treatment. It is not clear that knowing the baseline patient and disease characteristics are sufficient to achieve an improvement in cancer survivorship care over what is currently being done that would substantially decrease fatal stroke incidence.
- The observation of the substantially increased SMR for younger patients is interesting. However, might this be because they are enriched for brain tumors, which are known to have a higher incidence of stroke due to brain surgery. Can this finding be further supported with other data? Is this a consequence of chemo-aging that results in higher incidence of stroke (a disease of older age)?

Reviewer #2 (Remarks to the Author):

This is an interesting analysis of stroke risk after cancer in a large dataset.

The focus of the paper should be on the SRMs compared to the general population.

This is now given in Table 1 for all cancers, and in Figure 1 for selected cancer sites. Given the size of the dataset, additional information could be added for other cancer sites in tabular or figure form.

The so called plurality is essentially function on the different incidence of various cancers, thus it is less relevant to me.

As a minor point, the term contemporary analysis is unclear to me.

Reviewer #3 (Remarks to the Author):

This is an interesting SEER analysis comprehensively studying the standardized mortality ratios of fatal strokes in patients with multiple types of cancer of multiple ages with multiple follow-up times. The methodology is robust and thorough, and the details of the analysis and results are described and presented well.

I have one question regarding Figure 1: not all primary sites appear to be shown here. Is it because some did not have a high enough SMR? I am particularly interested in H+N cancers like oral cavity/pharynx and larynx, given conflicting studies in the past about whether or not patients who receive surgery or radiation therapy for these disease sites have an increased risk of (fatal) cerebrovascular events. Perhaps an additional table with the SMR for all disease sites (not necessarily subgrouping by follow-up time or age) would be helpful in understanding which disease sites have high, intermediate, or low risk of fatal CVAs compared to the general population.

Reviewer #1 (Remarks to the Author):

The objective of trying to identify cancer patients at risk for stroke is an important one. However, the limitations of the data source considerably limit the ability to glean useful information along this objective. As noted by the authors, the SEER database is limited in terms of the information it contains. The important information missing are patient co-morbidities at baseline. It is likely that there are important overlapping risk factors for stroke and cancer such as diet, exercise, diabetes, and smoking. Hence, it is not clear that a comparison to the general population is that informative but what would be more informative is a comparison of stroke incidence of cancer patients to a population enriched for stroke risk factors. This would allow the identification of risk factors that are unique to cancer patients outside of the known stroke risk factors. In addition, the variables in the SEER database are relatively crude and primarily limited to baseline patient and disease characteristics and first treatment course for the cancer. This severely limits being able to identify the cancer patient subgroups that would benefit the most for heightened stroke prevention measures. This is important given that only about 1% of the cancer patients suffered a fatal stroke. Given this low incidence, it does not necessarily make sense to implement something for the entire population given only a small number of patients may benefit.

AUTHOR RESPONSE: Thank you for your helpful suggestions. We greatly value this feedback, which significantly improves our research.

In this work, our objectives are to identify cancer patients at highest risk of fatal stroke compared to (1) the general population, and (2) other cancer patients. To achieve these goals, we need a single database that contains data on the entire population (both cancer and non-cancer patients), capturing information over a long period of time (i.e. decades), since death from stroke may occur many years after diagnosis. To the knowledge of the authors the current analysis with SEER database is the only one that can achieve these goals, and it is the largest analysis to achieve both goals.

There are two points we would like to address with respect to our reviewer's comment:

First, we agree that it would be ideal to identify overlapping risk factors for stroke such as diet, exercise, diabetes, and smoking. This type of analysis would expand on objective number 2, at the sacrifice of objective number 1; an analysis like this could be performed with a hospital database, but it would be much smaller in size, have shorter follow up, include fewer cancer types, and draw less meaningful conclusions. We now provide a table of all prior studies on the topic of stroke among cancer patients, below and in the discussion section. These studies have been performed to identify risk factors that are unique to cancer patients outside of the known stroke risk factors (now **Table 3**). Notably, in these studies, the number of patients is relatively small (typically <100 vs >80,000 in the current work), the types of patients included are much more limited (the other studies only focus on lung, prostate, colorectal, or breast cancer patients; ours includes all cancers among all ages), and the conclusions of other works are therefore limited. Thus, while we agree that identification of other risk factors would be ideal, there is currently no data set in the world that can achieve this goal on the same order of magnitude as the current analysis.

Table 3. Literature review on the analyses of stroke among cancer patients.

Study	Type of article / location	n	Principal cancers included	Follow up time	Findings
Zaorsky et al (Current study)	USA	7,529,481 cancer patients, 80,513 died of stroke (larger than all other articles below combined)	All. Notably, includes cancers of head and neck, heme system, pediatrics, GI, brain (which are typically not included in studies below); provides risk vs general and cancer populations	>20 years for cohort follow up, > 260,000 person years at risk for standardized incidence ratios. Median time of death for fatal stroke 5 months for brain tumor patients vs 52 months for non-brain patients.	Brain and GI cancer patients had the highest SMRs (> 2-5) through the follow up period. Among those diagnosed at < 40 years of age, plurality of strokes occurs in patients treated for brain tumors and lymphomas; if > 40, from cancers of the prostate, breast, and colorectum. For almost all cancer survivors, the risk of stroke increases with time.
Kim et al, 2010 ¹	Korea, 6 centers	161 cancer patients who had stroke	Mostly lung, gastric, colorectal	None, no survival analyses possible	Patients are at risk for cryptogenic and conventional stroke. D-dimer levels are higher in stroke patients (odds ratio 10-11).
Jagsi et al, 2006 ²	Ann Arbor, Michigan, USA	820 early breast cancer patients, 35 had stroke	Early breast only	6.8 years	SMR of stroke is 1.7-2.8 among breast cancer patients. Hypertension and age are predictors.
Stefan et al, 2009 ³	Vienna, Austria	1274 stroke patients, 12% of these had cancer	Mostly breast, prostate, colorectal	None, no survival analyses possible	Cerebrovascular risk factors do not significantly vary between cancer and non-cancer patients.
Zhang et al, 2007 ⁴	Australia	69 stroke patients with cancer at 1 hospital	Mostly prostate, lymphoma	None, no survival analyses possible	In cancer patients, trend toward higher risk of intracerebral hemorrhage and higher partial thromboplastin

					time.
Lindvig et al, 1990 ⁵	Denmark	113,732 stroke patients, 5,151 had cancer	Mostly lung	2.4 years	Overall, more cancer was expected than observed. No risk factor between stroke and gastric cancer
Chaturvedi et al, 1994 ⁶	U Massachusetts, MA, USA	33 patients with cancer who had stroke, representing 3.5% of admissions to hospital	Mostly GYN, genitourinary, gastrointestinal	9 months	Recurrent cerebral ischemic events were noted in only 6% of patients
Cestari et al, 2004 ⁷	NYC, NY, USA	96 patients with cancer who had stroke	Mostly lung, breast, prostate	After stroke median survival was 5 months	54% of patients had embolic strokes, partially due to hypercoagulability, with 11/12 patients having elevated D-dimer levels

Second, we agree that the incidence of *fatal* stroke among cancer patients is relatively low at 1%. In other analyses (**Table 3**), the rate of *non-fatal* strokes has been reported at ~5% among cancer patients, and even strokes that do not cause death may be debilitating. However, since there is a gradient with stroke diagnoses (e.g. transient ischemic attacks that leave no symptoms), we focus on only the most significant strokes (i.e. those that cause death), and we believe the results may be extrapolated to other cancer patients. We have integrated this in the discussion section.

Third, we believe that the results of the current analysis may be used to shape future policies and guidelines for stroke prevention for several reasons:

(a) In the current analysis, a total of 7,529,481 cancer patients were included in our analysis; of these, 80,513 (1.1%) died of a stroke. We note that among the patients who died of stroke, the rates of death are not evenly distributed, and there are certain patient and treatment related factors that place patients at an exceedingly high risk of fatal stroke. In the current work, patients with cancer of the prostate, breast, and colorectum contribute to the plurality of cancer patients dying of fatal stroke. Brain and gastrointestinal cancer patients had the highest SMRs (> 2-5) through the follow up period. Among those diagnosed at < 40 years of age, the plurality of strokes occurs in patients treated for brain tumors and lymphomas; if > 40, from cancers of the prostate, breast, and colorectum. For almost all cancers survivors, the risk of stroke increases with time.

(b) The relative risk of death from fatal stroke (as measured with the standardized mortality ratio, SMR) for 1992 - 2002 vs 2005 - 2015, is 1.58 (95% CI 1.55, 1.61) vs 2.95 (95% 2.91, 2.99). We hypothesize that the SMRs from more recent years may be higher than those seen in the 1990s because in the United States there has been a trend to detect and treat more low risk cancers (e.g. prostate, breast, colorectal); these patients likely will never die of their primary cancers and are therefore at higher risk to die of other causes.

Please note that the data for point (b) is from additional analyses that have been performed, as requested by one of the reviewers, below.

Thus, our results and conclusions have never been shown in any other work; juxtaposing the current analysis to other works highlights how this is the first comprehensive analysis on stroke among cancer patients (**Table 3**). We agree with the reviewer that implementation of a policy for the entire cancer population may not be the appropriate response. Instead, the current seminal analysis can be used in the recommendations of organizations to identify subgroups at risk of stroke and recommend appropriate screening and follow-up recommendations. We have clarified our Discussion section to state:

We encourage individual guideline and survivorship committees to incorporate these data into their stroke prevention statements. We recommend that providers follow the evolving guidelines for monitoring distress and stroke prevention from the National Comprehensive Cancer Network, American Heart Association and American Stroke Association.⁸⁻¹⁰

Specific items for consideration:

- Given that cancer is a given risk factor for cancer, is there any evidence that implementing stroke risk identification/prevention beyond what is currently being done would reduce fatal stroke incidence in this patient population? Specifically, is there evidence that cancer patients who are not at higher risk for stroke given other co-morbidities and lifestyle choices are at higher risk of stroke? The concern is that the higher observed SMR may reflect a population that has a higher burden of risk factors for stroke rather than a mechanism of the cancer.

AUTHOR RESPONSE: Thank you for your helpful suggestions. In current guidelines from national and international organizations, the recommendations regarding stroke prevention in cancer patients are limited, or absent entirely. In our Introduction, have clarified to state:

The American Heart Association and American Stroke Association provide guidelines for the prevention of stroke in patients with stroke and transient ischemic attack⁸ and for early management of patients with acute ischemic stroke.⁹ The National Comprehensive Cancer Network provides survivorship guidelines after therapy for cancer, with the goal of preventing long-term morbidity and mortality.¹⁰ As of 2019,

these organizations offer relatively limited guidelines for stroke prevention, identification, or management specifically in cancer patients. Thus, there is currently no resource to assist clinicians, including primary care physicians, oncologists, neurologists, neurosurgeons, and cardiologists, in identifying cancer patients at highest risk of stroke.

Second, with respect to the second question: Yes, there is evidence that cancer patients who are not at higher risk for stroke given other co-morbidities are at higher risk of stroke. We have performed additional analyses that plot the SMRs vs patient age at diagnosis, and we show that the SMRs are a function of both the cancer type and the age at diagnosis, and not because of higher burden of risk factors for stroke. In fact, younger patients, who typically have no comorbidities, are those at highest risk of stroke vs the general population.

In the **Figure 2** (and its Legend) we state:

The y-axis depicts the SMR, and the x-axis depicts the age at diagnosis with cancer. SMRs compare the risk of death from stroke among a cancer subsite vs the general population, adjusted sex and race, within a particular age subgroup. Cancers are shown in different colors; for the purposes of this figure, key cancers were selected because of their high incidence and prevalence overall (e.g. prostate, breast, colorectum)

and because of their relatively high incidence in pediatric populations (e.g. brain, leukemia); this was done so that SMRs between adult and pediatric populations may be juxtaposed. For pediatric patients, the population is enriched with brain tumors, and these contribute to the majority of person years at risk. Children diagnosed with brain tumors are at an exceedingly high risk to die of stroke for the remainder of their life (SMRs >100, p-values < 0.001). Adolescents and young adults who are diagnosed with leukemia are similarly at a high risk of death from fatal stroke (SMRs >100, p-values < 0.001). Since most cancers are diagnosed in adults and the elderly, SMRs for the majority of other cancers are not plotted until age 40 and over. In general, the younger a patient's age of diagnosis, the higher the SMR that the patient will die of stroke through their life.

Finally, in the Discussion section, we also clarify novel potential screening strategies, which are mentioned in **Table 3**:

Similar to previous analyses, we found that lung, prostate, breast, and colorectal patients experience the plurality of strokes. Although the current analysis does not include patient comorbidities or biomarkers, other studies suggest that D-dimer levels and classic risk factors for stroke (e.g. hypertension) put patients at greatest risk.

• The authors have mentioned several limitations that make it difficult to interpret these results. These include (1) the changes in stroke diagnosis and treatment and cancer diagnosis and treatment over the 23 years included in this analysis (how do the authors explain the increasing SMR as the date of diagnosis is more recent?), (2) the inability to distinguish between hemorrhagic and ischemic stroke (given some cancer treatments increase the risk of major bleeds), (3) and confounding by lifestyle choices that may place a patient at increased risk for both stroke and cancer. How do the authors envision their findings being able to guide physicians in the care of cancer survivors beyond what they are already doing?

AUTHOR RESPONSE: Thank you for your helpful suggestions. We address each point separately. With respect to each point we now state in the Discussion section:

(1) The changes in stroke diagnosis and treatment and cancer diagnosis and treatment have changed from 1992 to 2015. There may be some variability in the relative risk of fatal stroke among patients diagnosed in earlier years vs later years. Thus, we have performed additional analyses to address this concern, and we note that the relative risk of death from fatal stroke (as measured with the standardized mortality ratio, SMR) for 1992 - 2002 vs 2005 - 2015, is 1.58 (95% CI 1.55, 1.61) vs 2.95 (95% 2.91, 2.99). We hypothesize that the SMRs from more recent years may be higher than those seen in the 1990s because in the United States there has been a trend to detect and treat more low risk cancers (e.g. prostate, breast, colorectal); these patients likely will never die of their primary cancers and are therefore at higher risk to die of other causes.

We have added this change to the Discussion section.

We provide the full dataset for these time periods. (**Supplementary Data Set 6**).

(2 and 3) While SEER database was ideal for this analysis, it has some limitations. SEER contains basic information regarding diagnosis, cause of death, and first treatment type. It does not contain information regarding stroke subtype, comorbidities that may increase a patient's risk of stroke (e.g. smoking, hypertension), biomarkers (e.g. prothrombin time, D-dimer levels), or the full extent of their treatment. Notably, no other databases published to date contain these covariates (**Table 3**), and a nationally representative database containing this much information does not exist. We recognize these limitations and do not attempt to extrapolate our findings to specific subpopulations beyond the variables included in the analysis.

To mitigate these concerns, we have now performed additional analyses that plot the SMRs vs patient age at diagnosis, and we show that the SMRs are a function of both the cancer type and the age at diagnosis, and not because of higher burden of risk factors for stroke. In fact, younger patients, who typically have no comorbidities, are those at highest risk of stroke vs the general population.

These findings are valuable in better understanding stroke risk in cancer patients.

- It seems like there is a competing risks issue. In particular, patients who die early due to their cancer would no longer be at risk for a fatal stroke. Has this been considered?

AUTHOR RESPONSE: Thank you for your helpful suggestions. Yes, we consider competing risks in the analyses. We now state in the methods section:

In the analyses for both objective 1 and objective 2 of this manuscript, patients who die of their cancer are no longer at risk for a fatal stroke, and this is considered in the standardized mortality ratios, odds ratios, and hazard ratios.

- The increase in stroke in brain tumor patients may be the result of an increase in stroke due to brain surgery (a known risk factor). Is there any way to determine whether brain cancer patients had a shorter time to fatal stroke versus other types of cancer patients (i.e. has some type of time to event analysis been performed to try to understand this?)

AUTHOR RESPONSE: Thank you for your helpful suggestions. We have performed additional analyses as requested. We now state in the results section section:

Finally, we performed a subgroup analysis of patients with only brain tumors, to investigate if the risk of stroke was substantially higher in these patients. Among patients with brain tumors, there were 525 events of 99,434 patients, with a median time to stroke

of 5 months; in contrast, among other patients, there were 79,988 events, out of 743,0051 patients, with a median time to fatal stroke of 52 months. In the adjusted Cox regression, the hazard ratio of fatal stroke in brain tumor patients vs all others was 3.085 (95% CI 2.824, 3.369, $p < 0.0001$).

- How reliable are causes of death in the SEER database? What is the validity of this outcome?

AUTHOR RESPONSE: Thank you for your helpful comment. The data in SEER are managed by the National Cancer Institute and are very reliable. In the Supplemental Information section, we discuss the quality assurance and definitions as provided by the National Cancer Institute, which oversees the SEER database.

Quality assurance and completeness

SEER undergoes quality assurance using systematic, standardized, and periodic data collection procedure for all defined members of a defined cohort is performed to avoid surveillance bias.¹¹ The case-finding audits are performed by a qualified member from each SEER registry under the direction of members of the National Cancer Institute. Auditors create an abstract that contains the primary site and the case finding source.¹² When performing audits, SEER adheres to two basic principles: auditing high quantity and high risk data. High quantity refers to disease sites that have the highest incidence and prevalence (e.g. breast, prostate, lung, colon); as well facilities that contribute the greatest percent of cases to the central database. Additionally, pathology laboratories are selected to review tissue from patients not seen at that hospital. High risk refers to cases that are likely to be miscoded (e.g. head and neck, hematopoietic diseases); compliance to new rules; and newly-reportable diseases.

Defining the cause of death

Mortality codes in SEER are assigned from death certificates, completed by the doctor caring for the patient at the time of demise. There is no single best method for calculating survival from cancer in the SEER program.¹³ Different methods can give different outcomes, but for most variants considered the differences are small. For stroke, there is likely little discrepancy in the cause of death, as compared to a cause of death like heart disease, which may be caused by the cancer treatment, underlying heart disease, or a combination of both.

- It is noted that stroke rates differ by disease site, age, gender, marital status, and time after diagnosis. Are any of these factors new and would any of these impactfully influence how cancer survivors should be managed differently than what is already being done? It seems like these are

fairly crude measures and would not be able to tailor cancer survivorship plans to result in meaningful decreases in fatal stroke incidence.

AUTHOR RESPONSE: Thank you for your helpful comment. We agree that stroke rates differ depending on these covariates. Our results and conclusions have never been shown in any previous work; juxtaposing the current analysis to other works highlights how this is the first analysis of its kind (**Table 3**). The only other databases focusing on cancer patients who experience stroke typically have < 100 patients, follow up of a few years, and only a few types of cancer patients. We believe that the current seminal paper can be used in their recommendations to identify subgroups at risk of stroke and recommend appropriate screening and follow-up recommendations.

In our Discussion section we now state:

We present a contemporary analysis of risk of fatal stroke among more than 7.5 million cancer patients and report that stroke risk varies as a function of disease site, age, gender, marital status, and time after diagnosis. The risk of stroke among cancer patients is twice times that of the general population and rises with longer follow up time. The relative risk of fatal stroke, vs the general population, is highest in those with cancers of the brain and gastrointestinal tract. The plurality of strokes occurs in patients >40 years of age with cancers of the prostate, breast, and colorectum. Patients of any age diagnosed with brain tumors and lymphomas are at risk for stroke throughout life.

Most cancer patients now die of non-cancer causes.¹⁴ The results of the current work suggest that stroke-prevention strategies may be aimed at patients treated for brain tumors and lymphomas (particularly children) and older patients (i.e. > 40 year-olds) diagnosed with cancers of the prostate, breast, and colorectum. Though relatively less common, patients with cancers of the gastrointestinal tract (especially the pancreas, liver, esophagus) are at a relatively high risk to die of stroke at any time after diagnosis (SMRs 3-10). We encourage individual guideline and survivorship committees to incorporate these data into their stroke prevention statements. We recommend that providers follow the evolving guidelines for monitoring distress and stroke prevention from the National Comprehensive Cancer Network, American Heart Association and American Stroke Association.⁸⁻¹⁰

- A real concern is that one biggest risk factor for subsequent stroke may be the type of treatment a patient received. This has not been analyzed in this study. Have the authors tried to analyze this? Even if possible, there is an additional concern that the patients only have data available for their first line of treatment. Longer cancer survivors are more likely to have undergone several different treatments, which may partially explain that the patients who have survived longer (i.e. a longer time from diagnosis) have higher SMRs in general. It seems that in order to more finely

identify cancer patients at higher risk of fatal stroke, the most important variables would be stroke risk factors (so the analysis can adjust for these to find new factors) and patient treatment. There is also concern that once diagnosed with cancer, patient lifestyles may change in a way that places them at higher risk of stroke. These changes may be disease site specific as a consequence of morbidities associated with the cancer and its treatment. It is not clear that knowing the baseline patient and disease characteristics are sufficient to achieve an improvement in cancer survivorship care over what is currently being done that would substantially decrease fatal stroke incidence.

AUTHOR RESPONSE: Thank you for your comments. A database that contains information regarding cancer site, treatment information, comorbidities, stroke subtypes, with enough information to compare to the general population does not exist. Nonetheless, the current analysis is by far the largest of its kind, and it addresses two objectives that have never been addressed before in any database. We acknowledge the limitations that exist when working with SEER data, and have addressed these in our comments in the Discussion:

While SEER data was beneficial to use for this sort of analysis, it is not without limitations. SEER contains basic information regarding diagnosis, cause of death, and first treatment type. It does not contain information regarding stroke subtype, comorbidities that may increase a patient's risk of stroke (e.g. smoking, hypertension), biomarkers (e.g. prothrombin time, D-dimer levels), or the full extent of their treatment. Notably, no other databases published to date can do any of these (**Table 3**). To the best of our knowledge, a nationally representative database containing this much information does not exist. We recognize these limitations and do not attempt to extrapolate our findings to specific subpopulations beyond the variables included in the analysis. Our findings are valuable in better understanding stroke risk in cancer patients.

Additionally, in the Methods section we state:

The overview and limitations of the database and the methods are described in **Supplementary Data Set 1**.^{11-13,15} SEER is a network of population-based incident tumor registries from geographically distinct regions in the US, covering 28% of the US population, including incidence, survival, and surgical treatment.^{16,17} For the current analysis, the SEER 18 registry was used. The SEER registry includes data on sex, age at diagnosis, race, marital status, and year of diagnosis. SEER does not code comorbidities, performance status, surgical pathology, doses, radiotherapy use, and chemotherapy agents. SEER*Stat 8.2.1 was used for analysis.¹⁶

- The observation of the substantially increased SMR for younger patients is interesting. However, might this be because they are enriched for brain tumors, which are known to have a higher incidence of stroke due to brain surgery. Can this finding be further supported with other

data? Is this a consequence of chemo-aging that results in higher incidence of stroke (a disease of older age)?

AUTHOR RESPONSE: Thank you for your comments. Per your request, we have performed additional analyses, and we now plot the SMRs vs the age group of diagnosis in **Figure 2**. In the legend we state:

Figure 2. Standardized mortality ratios (SMRs) of fatal stroke as a function of age of diagnosis.

The y-axis depicts the SMR, and the x-axis depicts the age at diagnosis with cancer. SMRs compare the risk of death from stroke among a cancer subsite vs the general population, adjusted sex and race, within a particular age subgroup. Cancers are shown in different colors; for the purposes of this figure, key cancers were selected because of their high incidence and prevalence overall (e.g. prostate, breast, colorectum) and because of their relatively high incidence in pediatric populations (e.g. brain, leukemia); this was done so that SMRs between adult and pediatric populations may be juxtaposed. For pediatric patients, the population is enriched with brain tumors, and these contribute to the majority of person years at risk. Children diagnosed with brain tumors are at an exceedingly high risk to die of stroke for the remainder of their life (SMRs >100, p-values < 0.001). Adolescents and young adults who are diagnosed with leukemia are similarly at a high risk of death from fatal stroke (SMRs >100, p-values < 0.001). Since most cancers are diagnosed in adults and the elderly, SMRs for the majority of other cancers are not plotted until age 40 and over. In general, the younger a patient's age of diagnosis, the higher the SMR that the patient will die of stroke through their life.

For reference, we also provide **Supplementary Data Set 7**. Stroke standardized mortality ratios as a function of age of diagnosis. This file contains the raw data of the number of observed events, number of expected events, SMRs, 95% CIs, and the person years at risk for each age subgroup. We do not specifically comment on the impact of “chemo-aging” since SEER does not have detailed treatment information; the patients are at risk for stroke because of many factors, including tumor location, surgery, radiation, chemotherapy, subsequent comorbidities, lifestyle factors (e.g. smoking, exercise, diet).

Reviewer #2 (Remarks to the Author):

This is an interesting analysis of stroke risk after cancer in a large dataset.

The focus of the paper should be on the SRMs compared to the general population.

AUTHOR RESPONSE: Thank you for your supportive comments. We have made the SMRs the focus, and we have conducted additional analyses, as described above for reviewer #1.

This is now given in Table 1 for all cancers, and in Figure 1 for selected cancer sites. Given the size of the dataset, additional information could be added for other cancer sites in tabular or figure form.

AUTHOR RESPONSE: Thank you for your comments. We have made the SMRs the focus, and we have conducted additional analyses, as described above for reviewer #1.

The so called plurality is essentially function on the different incidence of various cancers, thus it is less relevant to me.

AUTHOR RESPONSE: Thank you for your comments. We agree that most strokes occur in cancers with a higher incidence. However, the relative risk of stroke vs the general population is very high in patients with brain tumors and gastrointestinal tumors; thus, we have kept the term plurality when discussing the relevant patient subpopulations.

As a minor point, the term contemporary analysis is unclear to me.

AUTHOR RESPONSE: Thank you for your comment. We have removed this term from the text.

Reviewer #3 (Remarks to the Author):

This is an interesting SEER analysis comprehensively studying the standardized mortality ratios of fatal strokes in patients with multiple types of cancer of multiple ages with multiple follow-up times. The methodology is robust and thorough, and the details of the analysis and results are described and presented well.

AUTHOR RESPONSE: Thank you for your supportive comments.

I have one question regarding Figure 1: not all primary sites appear to be shown here. Is it because some did not have a high enough SMR? I am particularly interested in H+N cancers like oral cavity/pharynx and larynx, given conflicting studies in the past about whether or not patients who receive surgery or radiation therapy for these disease sites have an increased risk of (fatal) cerebrovascular events. Perhaps an additional table with the SMR for all disease sites (not necessarily subgrouping by follow-up time or age) would be helpful in understanding which disease sites have high, intermediate, or low risk of fatal CVAs compared to the general population.

AUTHOR RESPONSE: Thank you for your supportive comments. We have performed additional analyses for head and neck cancer patients. and we have included “oral cavity and pharynx” as an overarching group in Figure 1. Our detailed analyses of all subsites are present in **Supplementary File 4**. We found that the SMRs among various head and neck subsites (e.g. oral cavity, oropharynx, larynx) were similar; thus, we felt it was more clear to readers to present one set of bar graphs (for <1 year, 1-5 years, and 5+ years) as “oral cavity and pharynx” as a whole, in Figure 1, rather than list every subsite. We also include a figure of the subsite SMRs in the supplement, so that interested readers may access the data.

In summary, we discuss the strengths and limitations of SEER, and we perform additional analyses, including sub-analyses of (1) younger patients and age at diagnosis, with respect to subsequent risk of stroke; (2) brain tumor patients only; and (3) head and neck tumor subsites. We also perform a thorough review of the literature regarding stroke among cancer patients, and we juxtapose the current analysis with previously published studies. The current work is by far the most comprehensive analysis of stroke in cancer patients. Please do not hesitate to contact us if you would like further analyses performed. Finally, we have submitted the editorial policy checklist, reporting summary, and source data files. Thank you again for the chance to revise our manuscript. If you request other edits or revisions, please do not hesitate to contact us.

Sincerely,
Nicholas G Zaorsky MD
Assistant Professor (Tenure-track), Department of Radiation Oncology, Penn State Cancer
Institute, Hershey, PA
Assistant Professor, Department of Public Health Sciences, Penn State Health Milton S. Hershey
Medical Center, Hershey, PA
Physician-leader, Radiation Oncology Genitourinary Cancer Program
Physician-leader, Radiation Oncology Research Program
500 University Drive
Hershey, PA 17033
USA. Tel: +1-717-531-8024
Fax: +1-717-531-0446
E-mail: nicholaszaorsky@gmail.com; nzaorsky@pennstatehealth.psu.edu

REFERENCES

- 1 Kim, S. G. *et al.* Ischemic stroke in cancer patients with and without conventional mechanisms: a multicenter study in Korea. *Stroke* 41, 798-801, doi:10.1161/STROKEAHA.109.571356 (2010).
- 2 Jagsi, R., Griffith, K. A., Koelling, T., Roberts, R. & Pierce, L. J. Stroke rates and risk factors in patients treated with radiation therapy for early-stage breast cancer. *J. Clin. Oncol.* 24, 2779-2785, doi:10.1200/JCO.2005.04.0014 (2006).
- 3 Stefan, O., Vera, N., Otto, B., Heinz, L. & Wolfgang, G. Stroke in cancer patients: a risk factor analysis. *J. Neurooncol.* 94, 221-226, doi:10.1007/s11060-009-9818-3 (2009).
- 4 Zhang, Y. Y. *et al.* Risk factor, pattern, etiology and outcome in ischemic stroke patients with cancer: a nested case-control study. *Cerebrovasc. Dis.* 23, 181-187, doi:10.1159/000097639 (2007).
- 5 Lindvig, K., Moller, H., Mosbech, J. & Jensen, O. M. The pattern of cancer in a large cohort of stroke patients. *Int. J. Epidemiol.* 19, 498-504 (1990).
- 6 Chaturvedi, S., Ansell, J. & Recht, L. Should cerebral ischemic events in cancer patients be considered a manifestation of hypercoagulability? *Stroke* 25, 1215-1218 (1994).
- 7 Cestari, D. M., Weine, D. M., Panageas, K. S., Segal, A. Z. & DeAngelis, L. M. Stroke in patients with cancer: incidence and etiology. *Neurology* 62, 2025-2030 (2004).
- 8 Kernan, W. N. *et al.* Guidelines for the prevention of stroke in patients with stroke and transient ischemic attack: a guideline for healthcare professionals from the American Heart Association/American Stroke Association. *Stroke* 45, 2160-2236, doi:10.1161/STR.0000000000000024 (2014).
- 9 Powers, W. J. *et al.* 2018 Guidelines for the Early Management of Patients With Acute Ischemic Stroke: A Guideline for Healthcare Professionals From the American Heart Association/American Stroke Association. *Stroke* 49, e46-e110, doi:10.1161/STR.0000000000000158 (2018).
- 10 Denlinger, C. S. *et al.* Survivorship, Version 2.2018, NCCN Clinical Practice Guidelines in Oncology. *J. Natl. Compr. Canc. Netw.* 16, 1216-1247, doi:10.6004/jnccn.2018.0078 (2018).
- 11 Park, H. S., Lloyd, S., Decker, R. H., Wilson, L. D. & Yu, J. B. Overview of the Surveillance, Epidemiology, and End Results database: evolution, data variables, and quality assurance. *Curr. Probl. Cancer* 36, 183-190, doi:10.1016/j.currprobcancer.2012.03.007 (2012).
- 12 National Cancer Institute. Casefinding Studies - SEER Quality Improvement., <<http://seer.cancer.gov/qi/tools/casefinding.html>> (2016).
- 13 Boer, R. *et al.* (Statistical Research and Applications Branch, NCI, Bethesda, MD).
- 14 Zaorsky, N. G. *et al.* Causes of death among cancer patients. *Ann. Oncol.* 28, 400-407, doi:10.1093/annonc/mdw604 (2017).
- 15 Zaorsky, N. G. *et al.* Suicide among cancer patients. *Nat. Commun.* 10, 207, doi:10.1038/s41467-018-08170-1 (2019).
- 16 Surveillance Research Program, National Cancer Institute SEER*Stat software version 8.2.1., <www.seer.cancer.gov/seerstat> (2016).
- 17 Warren, J. L., Klabunde, C. N., Schrag, D., Bach, P. B. & Riley, G. F. Overview of the SEER-Medicare data: content, research applications, and generalizability to the United States elderly population. *Med. Care* 40, IV-3-18 (2002).

REVIEWERS' COMMENTS:

Reviewer #1 (Remarks to the Author):

The authors were responsive to review comments and have done the best they can with the limitations of using this dataset to address the question at hand.